# Diminazene Aceturate Reduces Angiotensin II Constriction and Interacts with the Spike Protein of Severe Acute Respiratory Syndrome Coronavirus 2

**DOI:** 10.3390/biomedicines10071731

**Published:** 2022-07-18

**Authors:** John M. Matsoukas, Laura Kate Gadanec, Anthony Zulli, Vasso Apostolopoulos, Konstantinos Kelaidonis, Irene Ligielli, Kalliopi Moschovou, Nikitas Georgiou, Panagiotis Plotas, Christos T. Chasapis, Graham Moore, Harry Ridgway, Thomas Mavromoustakos

**Affiliations:** 1NewDrug PC, Patras Science Park, 26500 Patras, Greece; k.kelaidonis@gmail.com; 2Institute for Health and Sport, Victoria University, Melbourne, VIC 3030, Australia; laura.gadanec@live.vu.edu.au (L.K.G.); anthony.zulli@vu.edu.au (A.Z.); vasso.apostolopoulos@vu.edu.au (V.A.); 3Department of Physiology and Pharmacology, Cumming School of Medicine, University of Calgary, Calgary, AB T2N 4N1, Canada; 4Immunology Program, Australian Institute for Musculoskeletal Science, Melbourne, VIC 3021, Australia; 5Department of Chemistry National and Kapodistrian, University of Athens, Zographou, 15784 Athens, Greece; eir.ligielli@gmail.com (I.L.); kmoschovou@chem.uoa.gr (K.M.); nikitage@chem.uoa.gr (N.G.); 6Laboratory of Primary Health Care, School of Health Rehabilitation Sciences, University of Patras, 26504 Patras, Greece; panagiotisplotasmed@gmail.com; 7NMR Facility, Instrumental Analysis Laboratory, School of Natural Sciences, University of Patras, 26504 Patras, Greece; cchasapis@upatras.gr; 8Institute of Chemical Engineering Sciences, Foundation for Research and Technology, Hellas (FORTH/ICE-HT), 26504 Patras, Greece; 9Pepmetics Incorporated, 772 Murphy Pace, Victoria, BC V8Y 3H4, Canada; mooregj@shaw.ca; 10Institute for Sustainable Industries and Liveable Cities, Victoria University, Melbourne, VIC 8001, Australia; ridgway@vtc.net; 11AquaMem Consultants, Rodeo, NM 88056, USA

**Keywords:** angiotensin-converting enzyme 2, angiotensin II, angiotensin type 1 receptor, coronavirus 2019, diminazene aceturate, sartans, severe acute respiratory syndrome coronavirus 2, SARS-CoV-2, COVID-19, drug repurposing, DIZE, ACE2

## Abstract

Diminazene aceturate (DIZE) is a putative angiotensin-converting enzyme 2 (ACE2) activator and angiotensin type 1 receptor antagonist (AT_1_R). Its simple chemical structure possesses a negatively charged triazene segment that is homologous to the tetrazole of angiotensin receptor blockers (ARB), which explains its AT_1_R antagonistic activity. Additionally, the activation of ACE2 by DIZE converts the toxic octapeptide angiotensin II (AngII) to the heptapeptides angiotensin 1–7 and alamandine, which promote vasodilation and maintains homeostatic balance. Due to DIZE’s protective cardiovascular and pulmonary effects and its ability to target ACE2 (the predominant receptor utilized by severe acute respiratory syndrome coronavirus 2 to enter host cells), it is a promising treatment for coronavirus 2019 (COVID-19). To determine DIZE’s ability to inhibit AngII constriction, in vitro isometric tension analysis was conducted on rabbit iliac arteries incubated with DIZE or candesartan and constricted with cumulative doses of AngII. In silico docking and ligand interaction studies were performed to investigate potential interactions between DIZE and other ARBs with AT_1_R and the spike protein/ACE2 complex. DIZE, similar to the other ARBs investigated, was able to abolish vasoconstriction in response to AngII and exhibited a binding affinity for the spike protein/ACE2 complex (PDB 6LZ6). These results support the potential of DIZE as a treatment for COVID-19.

## 1. Introduction

The angiotensin-converting enzyme 2 (ACE2) enzyme is the main receptor utilized by severe acute respiratory syndrome coronavirus 2 (SARS-CoV-2), the virus responsible for coronavirus 2019 (COVID-19), to enter host cells [1,2]. Due to SARS-CoV-2 interaction with ACE2, recent literature has suggested inhibiting ACE2 activity as a possible COVID-19 treatment [1,2]. The interaction between the SARS-CoV-2 virus spike protein receptor-binding domain (RBD) and the ACE2 cell surface protein is required for the viral infection of cells [1]. Patients with SARS-CoV-2 virus infection develop pulmonary edema and acute respiratory distress syndrome. The role of ACE2 in normal pulmonary and myocardial physiology is well-established, as ACE2 reduction leads to myocardial and pulmonary disease [3], and the loss of ACE2 function has been postulated to exacerbate the COVID-19 severity [1,4]. Therefore, it stands to reason that SARS-CoV-2 could be inhibiting ACE2 activity, leading to worsened outcomes [5]. A recent study has suggested that diminazene aceturate (DIZE), a putative ACE activator, may have the potential to prevent the production of a cytokine storm (an inflammatory response) produced in patients with SARS-CoV-2 [5].

DIZE is an antiparasitic agent approved by the US Food and Drug Administration for its use by veterinarians as a treatment for blood-borne protozoan infections (e.g., Trypanosoma and Babesia) in livestock and domestic animals [6]. DIZE exerts its protozoalcidal abilities by targeting and binding to circular and kinetoplast DNA, displaying a high affinity for adenine–thymine base pairs, which results in complete and irreparable DNA loss [7]. DIZE remains unlicensed for its use in humans; however, a long-term study involving 99 patients in the early stages of African trypanosomiasis disease reported no major toxicities [8]. Recent in silico and preclinical animal studies have demonstrated the novel ability of DIZE to activate ACE2 [9], which has shown beneficial and protective effects in multiple pathologies relating to the cardiovascular [10,11], pulmonary [12] and renal [13] systems (Figure 1). The activation of ACE2 results in the conversion of pathogenic angiotensin II (AngII) to the beneficial heptapeptides angiotensin 1–7 (Ang(1–7)) and alamandine [6], which comprise the counterregulatory alternative axis of the renin angiotensin system (RAS) [3]. It is well-documented that, upon ACE2 activation, alamandine and Ang(1–7) are generated to stimulate the Mas-related G-coupled protein receptor member (MRGD R) and Mas1 oncogene receptor (MAS R), respectively, to promote blood vessel relaxation [3] (Figure 1). In addition, the ability of DIZE to reduce the mean arterial blood pressure in rats with spontaneous hypertension suggests a direct association between DIZE and angiotensin type 1 receptor (AT_1_R) [11]. This interaction may be due to DIZE sharing a similar chemical structure to AngII and angiotensin receptor blockers (ARBs).

The simple chemical structure of DIZE is important for understanding its ARB mimicking effect, AT_1_R antagonistic behavior and ability to activate ACE2. Firstly, DIZE contains an acidic triazene proton, which is homologous to the tetrazole proton in losartan and other ARBs and is important for their pharmacodynamic role. All ARBs contain tetrazole or carboxyl groups as the essential AngII receptor negative charge feature for triggering activity. Triazene of DIZE and tetrazole of ARBs bearing negative charges mimic the C-terminal carboxyl group of the AngII obligatory requirement for activity. Secondly, a network spreading over the molecule is similar to that of the biphenyl tetrazole of losartan and other ARBs. The segment regioselectivity in DIZE and ARBs mimics the charge relay system (CRS) of AngII, which results in the tyrosinate pharmacophore that binds to the receptor. Lastly, two terminal amidine groups, which are strong proton acceptors, act similarly to the imidazole group of ARBs and the guanido group of arginine in AngII. These findings provide a fundamental basis that DIZE may be a target compound for the next generation of novel drugs against cardiovascular diseases [6,14]. However, despite evidence reporting the ability of DIZE to activate ACE2, a contradicting study demonstrated the failure of DIZE to enhance AngII degradation by recombinant ACE2 in in vitro and in vivo murine models, suggesting its beneficial effects may be independent of ACE2 [15].

DIZE has been postulated to exert its therapeutic abilities by modulating RAS, shifting it from a deleterious axis to one that promotes counterregulatory protective cardiovascular, renal, and pulmonary mechanisms. Animal studies have reported a positive relationship between DIZE treatment and augmented ACE2 activation and activity, as treatment with DIZE in rats has been shown to reduce blood pressure and prevent the progression of renovascular hypertension-induced cardiac hypertrophy via ACE2 and MAS R activation [16]. Interestingly, in renal tissue, increased glomeruli and whole kidney ACE2 and AT_2_R expression and reduced renal AngII levels and elevated Ang(1–7) levels have been observed in male Wistar rats with streptozotocin-induced diabetes [17]. However, a combination of DIZE and PD123319 (AT_2_R antagonist) reversed DIZE’s beneficial effects on the RAS components, suggesting that DIZE’s mechanism of action is not limited to ACE2 but, rather, affects different components of the ACE2/Ang(1–7)/AT_2_R counterregulatory RAS axis [17]. DIZE’s ability to target the traditional RAS axis is further supported by DIZE reducing ACE (60%) and AT_1_R (75%) and increasing the ACE2 (30%) mRNA levels in pulmonary tissue monocrotaline-challenged male Sprague–Dawley rats to induce pulmonary hypertension [12]. Additionally, treatment with DIZE in cultured human retinal pigment epithelial cells challenged with lipopolysaccharide had markedly increased mRNA and protein expression of Ang(1–7) and reduced AngII and AT_1_R [18]. Taken together, the results from these studies suggest that DIZE may be able to target different components of the RAS in a range of tissues, of which are greatly affected by SARS-CoV-2 infection, making it an ideal candidate to further investigate its antiviral ability to be used as a treatment for COVID-19.

Herein, we determined the ability of DIZE to behave as an ARB by blocking AngII-induced vasoconstriction in rabbit iliac arteries and its ability to disturb the spike protein receptor-binding domain/ACE2 complex.
Figure 1**Schematic diagram of the RAS and DIZE’s proposed involvement.** The RAS comprises two arms: the classical and alternative axes. In the classical axis, angiotensinogen produced by the liver is cleaved by renin to produce angiotensin I (AngI) [3]. AngI is then acted upon by angiotensin converting enzyme to produce AngII, which can associate with AT_1_R to increase the blood pressure, reduce nitric oxide production (the most potent endogenous vasodilator) and promote vasoconstriction. Additionally, AngII can bind to and activate the counterregulatory AT_4_R and AT_2_R, the latter of which is part of the alternative axis [3]. AngII can further be processed by aminopeptidase A (APA) to form angiotensin III (AngIII), which, in turn, can be cleaved by either APA or alanyl aminopeptidase to form angiotensin IV (AngIV). Both AngIII and AngIV are able to bind and activate AT_1_R, resulting in pathogenic cardiovascular consequences [3]. Alternatively, AngI can also be cleaved by ACE2 to produce angiotensin 1–9 (Ang(1–9)) or by neprilysin to generate Ang(1–7) [3]. Ang(1–9) and Ang(1–7) are both cardioprotective, and their association with AT_2_R or MAS R results in the opposing effects of AT_1_R. Ang(1–7) can also be produced from the ACE2 cleavage activity of AngII and can further be metabolized by aspartate decarboxylase (AD) into alamandine, which engages with MRGD R. AD is also able to cleave AngII to produce angiotensin A, which can be converted into alamandine by ACE2 [3]. DIZE is a putative ACE2 activator and AT_1_R inhibitor, which has been shown to directly cause vasodilation by increasing the nitric oxide production and MAS R activation, reduces the blood pressure and increases the cardioprotective peptides Ang(1–7) and AT_2_R [11,17,19]. Figure was made using biorender.com (accessed on 4 July 2022). Abbreviations: ACE, angiotensin-converting enzyme; ACE2, angiotensin-converting enzyme 2; AD, aspartate decarboxylase; APA, aminopeptidase A; APN, alanyl aminopeptidase; AT_1_R, angiotensin type 1 receptor; AT_2_R, angiotensin type 2 receptor; AT_4_R, angiotensin type 4 receptor; DIZE, diminazene aceturate; MAS R, Mas1 oncogene receptor; MRGD R, Mas-related G-coupled protein receptor member; NEP, neprilysin.
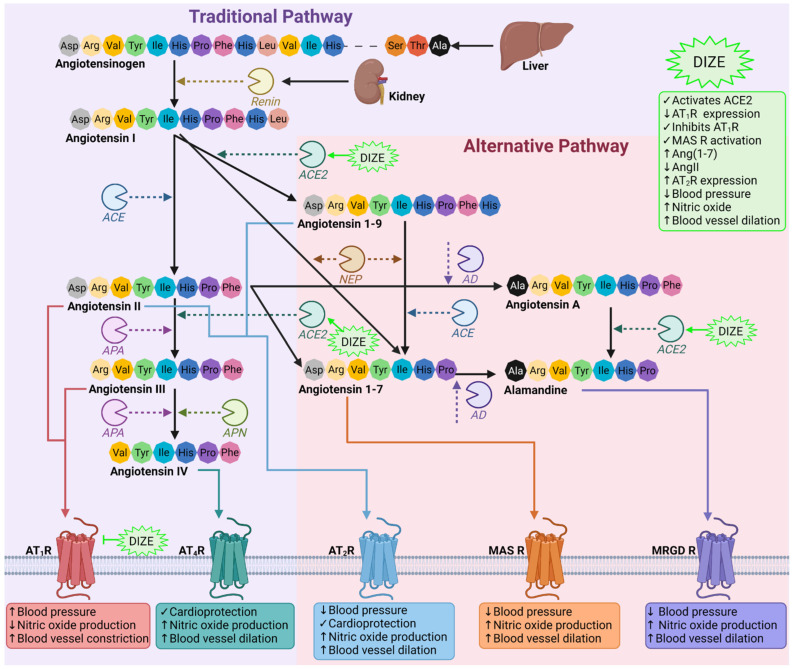



## 2. Materials and Methods

### 2.1. Animal In Vitro Experiments

#### 2.1.1. Materials

Candesartan (Cat#SML0245) was purchased from Sigma Aldrich (St. Louis, MO, USA), DIZE (Cat#18678) was purchased from Cayman Chemicals (Ann Arbor, MI, USA) and human AngII (Cat#51480) was purchased from Mimotopes (Melbourne, VIC, Australia).

#### 2.1.2. Animal Model and Ethics Approval

Male New Zealand White rabbits (*n* = 4) at 7 weeks of age were purchased from Flinders City University (Bedford Park, SA, Australia). The animals were housed individually at the Victoria University Werribee Campus Animal Facilities until 16 weeks of age. Upon arrival, animals were given a 7-day acclimatization period. Animals were kept on a 12-h day/night circadian rhythm cycle and were maintained at a constant temperature of 21 °C and relative humidity level between 40 and 70%. Food and water were supplied *ad libitum*. All experimental procedures were conducted in accordance with the National Health and Medical Research Council ‘*Australia Code of Practice for the Care and Use of Animals for Scientific Purposes*’ (8th edition, 2013, https://www.nhmrc.gov.au/about-us/publications/australian-code-care-and-use-animals-scientific-purposes accessed on 4 July 2022), and were approved by the Victoria University Animal Ethics Committee (VUAEC#17/013).

#### 2.1.3. Anesthesia, Euthanasia and Tissue Retrieval Protocol

Animals were first sedated using a subcutaneous injection of medetomidine (0.25 mg/kg) at the scruff of the neck. Animals were then anaesthetized using the inhalant anesthesia isoflurane (4%). Once correctly anesthetized (indicated by loss of pedal pain and corneal reflexes), an incision was made at the lower abdomen, and the subcutaneous tissue and lower abdominal muscles were dissected to locate the inferior vena cava. Animals were then exsanguinated via inferior vena cava perforation for approximately 3 min or until loss of color and dilation of pupils was observed. At this time, the diaphragm was dissected, signifying death of the animal. A T-tube was inserted into the distal aortic arch, allowing for flushing of the aorta, aortoiliac bifurcation, and iliacs with cold, oxygenated Krebs–Henseleit solution (Krebs). Both iliac arteries were retrieved from each animal, cleaned of fat and connective tissue under a light microscope and dissected into 2 to 3-mm rings in preparation for drug incubations and an isometric tension analysis.

#### 2.1.4. Drug Incubations and Isometric Tension Analysis

Iliac rings were immediately transferred sequentially into adjacent organ baths (Zultek Engineering, Melbourne, VIC, Australia) filled with 5 mL of Krebs. Organ baths were maintained at 37 °C and continuously bubbled with 95% carbogen. Rings were allowed to acclimatize for 15 min and were then mounted between two metal organ hooks attached to force displacement transducers, gently stretched to a resting tension of 0.5 g and allowed to equilibrate for 15 min. Rings were re-stretched, refreshed and equilibrated for a further 15 min. At this time, some iliac artery rings were: (a) left to rest for 10 min (control; *n* = 4), (b) incubated with candesartan (10^−7.0^ M) (a commercially available ARB) for 10 min to serve as an internal control (candesartan; *n* = 3) or (c) incubated with DIZE (10^−7.0^ M) for 10 min (DIZE; *n* = 3). To investigate the ability of DIZE to inhibit AngII-mediated constriction in comparison to candesartan, an AngII dose response (10^−12.0^ M–10^−5.0^ M) was then performed. Lastly, to determine the standardized maximal vasoconstriction abilities, the rings were allowed to return to the baseline tension, and a Krebs buffer was replaced with a high-potassium solution (125 mM).

#### 2.1.5. Statistical Analysis

GraphPad prism (Version 9.2.0, GraphPad Software Incorporated, San Diego, CA, USA) was utilized to analyze the isometric tension data. The significant *p*-value was set at *p* < 0.05, and a two-way ANOVA followed by Sidak’s multiple comparisons post hoc test was performed to determine the significance of the isometric tension analysis data. All data were represented as the mean ± standard error of the mean (SEM).

### 2.2. In Silico Methodology

#### 2.2.1. Ligand Preparation

The chemical structures of both DIZE and candesartan were constructed using the Chem3D 15.0 module of ChemOffice 15.0 and saved as mol2 files. The form of the molecules used were that of pH = 7.4 to simulate the relevant physiological environment [20] using Epik from the Maestro Schrödinger platform [21,22]. The structures were optimized using UCSF Chimera 1.14., where the charges of the molecular structures were assigned and their energies were minimized [23]. In detail, the ligands were energy-minimized using the steepest descent method for 1000 steps with a step size of 0.02. The hydrogens and Gasteiger charges were then assigned by employing an assisted model building with energy refinement (AMBER) force field [24,25]. The resulting files were saved in mol2 format [26]. In AutoDock 4.0, flexible torsions were assigned allowing 100 conformations [27,28,29], and the acyclic dihedral angles were allowed to rotate freely. The files were then saved in the pdbqt file format for further analysis [26].

#### 2.2.2. Molecular Docking

Molecular simulation studies of DIZE and candesartan into the protein targets were carried out using the open-source program AutoDock VINA included in Auto-Dock Tools 1.5.6 [27,28,29]. For this study, the crystal structures of the proteins were extracted by the Protein Data Bank. Two different sets of docking studies were conducted in total, where the first one included various SARS-CoV-2 proteins and the second a GPCR AT_1_R. The reason for those two categories was to examine how effective DIZE is as both an antiviral drug against SARS-CoV-2 and as an AT_1_R antagonist. These results were compared to candesartan, a known ARB [30]. The first set included the crystal structures of: the main protease (Mpro) of SARS-CoV-2 (PDB ID: 6LU7) [31], the RBD/ACE2 complex (Protein Data Bank, PDB ID: 6LZG) [31] and the SARS-CoV-2 S-protein at its open state (Protein Data Bank, PDB ID: 7DDN) [32].

The second molecular docking studies were carried out on the crystal structure of the human AT_1_R (PDB ID: 4YAY) [33]. The best docked poses, with both lower binding energies and stronger interaction patterns, were derived from the docking results and were visualized with PyMOL and the protein–ligand interaction profiler (PLIP; retrieved from http://www.pymol.org/pymol (accessed on 4 July 2022) [34,35].

The PLIP was also used to determine the interactions between the ligands and the proteins [25]. The X-ray crystal structures of the selected SARS-CoV-2 proteins were downloaded from Protein Data Bank as pdb files, and Open Babel software was used to convert the protein files into pdbqt types [36]. In all cases, the co-crystallized ligands were removed, thus the N3 Peptide-like/Inhibitor of Mpro (PDB ID: 6LU7), 2-acetamido-2-deoxy-beta-D-glucopyranose of spike protein (PDB ID: 6LZG) and 5,7-diethyl-1-{[2′-(1H-tetrazol-5-yl)biphenyl-4-yl]methyl}-3,4-dihydro-1,6-naphthyridin-2(1H)-one of the human AT_1_R (PDB ID: 4YAY). In the case of the spike protein at the open state (PDB ID: 7DDN), chains B and C were removed as well. This is because only the N501Y mutation of the alpha strain is located in the RBD region (residues 319–541); all other mutations are reported around the RBD region (i.e., HV-69-70del, Y-144del or after it, such as A570D, P681H, T716I, S982A and D1118H) [24]. The protein structures were refined for heteroatoms and water molecules to demarcate the active sites of the proteins. The hydrogen atoms and Kollman charges were added, and the nonpolar hydrogens were merged. In each receptor, the grid was set around its active site in order for site-specific docking to be performed [37].

The Lamarckian genetic algorithm (GA), which uses the AMBER force field to run the docking between the receptor and the ligand, was used for docking in combination with the grid-based energy evaluation method with the default parameters (GA: 2,500,000—energy evaluations and 150—population size). The program was run for a total number of 100 genetic algorithm runs. When using AutoDock VINA, the stability of the ligand/protein complex, which shows how efficiently can the ligand binds to the protein, is depicted with low energy. The lower the energy, the more efficient the ligand binds and is depicted by the increased number of hydrogen bonds, noncovalent Van Der Waals and hydrophobic interactions [38,39].

#### 2.2.3. Docking Parameters

For in silico SARS-CoV-2 protein experiments with the selected ligands, the active site of each receptor was targeted. For Mpro (PDB entry code: 6LU7), molecular docking was performed at its known active site, where the peptide-like inhibitor N3 is bound. The receptor grid box was generated by AutoGrid4 with grid box dimensions of 60 Å × 60 Å × 60 Å, with spacing of 0.375 Å centering around hotspot residues His41, Cys145, Ser144, Glu166 and Gln189 [31]. For the spike protein, which also includes its open state, the active center is in the RBD. For the complex spike protein, with ACE2 as its predominant receptor (PDB ID: 6LZG), the grid box was set with spacing of 0.420 Å and dimensions of 60 Å × 80 Å × 80 Å centering around residues Lys417, Leu455, Phe486, Asn487, Gln493 and Ser494, and for the open-state spike protein, the dimensions are 126 Å × 90 Å × 112 Å with spacing of 0.375 Å centering around residues Ser349, Asn450, Leu452, Phe490 and Leu492 [40,41,42,43,44]. The configuration file for the grid parameters was defined using 3D grid centers for the Mpro of SARS-CoV-2 with −10.450, 11.92 and 69.425 as the X-, Y- and Z-coordinates. For the spike protein of SARS-CoV-2 complexed with ACE2, the coordinates of the 3D grid centers are −36.176, 31.027 and −1.875 and, for the open-state spike protein, are 225.56, 263.908 and 289.107 as the X-, Y- and Z-coordinates, respectively [24].

In addition, the selected ligands were tested in AT_1_R (PDB ID: 4YAY). For this receptor, molecular docking was performed at its known active site, where the co-crystallized ligand was found. The grid box was set with spacing of 0.400 Å and dimensions of 70 Å × 65 Å × 70 Å. It was set in order to be centered to the residues Tyr35, Tyr92, Val108, Arg167 and Asp281. Respectively, the X-, Y- and Z-coordinates were set as −19.582, 14.763 and 37.654 [30,33].

#### 2.2.4. ADMET Calculations

Both compounds were sketched in Chem Draw and converted into SMILES in both web applications tools preADMET [45] and pKCSM [46]. Both programs were used to examine the drug-likeness (Lipinski’s Rule of Five [43] and Veber’s rule [44]). In addition, several toxicity parameters were examined via these tools. This procedure is very important for computational drug design because some potential biological compounds fail to reach clinical trials due to their unfavorable (ADME) parameters.

## 3. Results

### 3.1. In Vitro Animal Studies

#### DIZE Abolishes AngII-Mediated Constriction in Rabbit Iliac Arteries

Incubation with candesartan was able to abolish vasoconstriction responses to accumulative doses of AngII when compared to the controls at AngII (10^−9^ M) (control: 12.69 ± 1.82% vs. candesartan: 0.76 ± 0.31%, ** p* = 0.0345) to AngII (10^−6.5^ M) (control: 11.73 ± 5.26% vs. candesartan: 0.17 ± 7.6 × 10^−2^%, ** p* = 0.0279) (Figure 2 and Table 1). Strikingly, similar results were seen in the rings incubated with DIZE, as AngII-mediated vasoconstriction was also abolished from AngII (10^−9^ M) (control vs. DIZE: 0.20 ± 2.00 × 10^−3^, ** p* = 0.0267) to AngII (10^−6.5^ M) (control vs. DIZE: 0.72 ± 0.72%, ** p* = 0.0377) (Figure 2 and Table 1). Interestingly, no statistically significance differences were observed between DIZE and candesartan at any doses (Table 1). These results suggest the ability of DIZE to inhibit AngII mediated constriction which may be through interaction with AT_1_R, and is comparable to candesartan, a known and commercially available ARB.

### 3.2. In Silico Molecular Docking Analysis

Molecular docking was performed with AutoDock VINA [26]. DIZE and candesartan produced pdbqt files, which were converted to pdb, using Open babel [36] in order to be compatible with the visualization platforms. Then, the pdb files were visualized and analyzed by PyMOL and PLIP [34]. The most important issue in the probing of the conformational states of the ligands docked to receptors is the theoretical identification of the most significant conformation considering the most: (i) appropriate accommodations of ligands in the enzyme catalytic sites and (ii) promising interaction energy within the enzyme active site. Since the conformational changes occur in dozens of nanoseconds, the time scale could compromise the MD simulation viability, and there are various theoretical strategies in the literature that manage to increase the theoretical accuracy of virtual screening studies [27,28,29]. In this regard, the in silico experiments between the proteins and the ligands have been repeated in triplicate for each ligand and the average values for the binding affinity were recorded. The deviation of each compound was approximately ±0.5 the binding energy.

#### 3.2.1. Interactions of Ligands with Mpro

Mpro of SARS-CoV-2 (PDB ID: 6LU7) [31] displayed an excellent docking score of −10.2 kcal/mol with DIZE. It is comparable to the docking result of candesartan, which displays a score of −10.6 kcal/mol. DIZE formed eight hydrogen bonds, one with Met49, Tyr54, Phe140, Asn142, His172 and Gln189 and two with His164. Additionally, three hydrophobic interactions were also observed with His41, Met165 and Glu166. Apart from these, one π-stacking with His41 is formed. In the case of the candesartan, three hydrogen bonds can be observed, with His163, Met165 and His172. Apart from these are four hydrophobic bonds with Met165, Leu167, Pro168 and Gln192 and one π-stacking with His163 (Table 2).

#### 3.2.2. Interactions of Ligands with Spike Protein/ACE2

The effect of DIZE on the SARS-CoV-2 spike protein/ACE2 region was also investigated (PDB ID: 6LZG) [14]. DIZE displayed a higher binding affinity of −12.0 kcal/mol when compared to candesartan (−8.61 kcal/mol). DIZE formed seven hydrogen bonds: one with Ala396, Glu564 and Trp566 and two with Asn210 and Ser563. In addition, five hydrophobic interactions were formed with Leu91, Leu95, Val209, Val212 and Pro565. On the other hand, candesartan forms seven hydrogen bonds, one with Gln98, Gln102, Tyr 196 and Asp206 and three with Asn210. In addition, five hydrophobic interactions can be observed with Leu95, Asp206, Glu208, Val209 and Pro565, as well as one π-cation interaction with Lys562 and one salt bridge with the same amino acid (Table 3).

#### 3.2.3. Interactions of Ligands with Open State Spike Protein

The spike protein RBD of SARS-CoV-2 at the open state (PDB ID: 7DDN) [32] was also tested with the ligands. DIZE showed a lower binding energy (−6.30 kcal/mol) compared to candesartan, which had a binding energy of −7.91 kcal/mol. In detail, DIZE forms six hydrogen bonds: one with Tyr449, Glu484, Gln493 and with Ser494 and two with Leu492. In addition, there are four hydrophobic interactions: one with Tyr449, two with Leu452 and one with Phe490. On the other hand, candesartan forms three hydrogen bonds with Ser349, Asn450 and with Leu452. Candesartan also forms five hydrophobic interactions: two with Leu452 and one with each of the following residues: Asn450, Phe490 and Leu492 (Table 4).

#### 3.2.4. Interactions of Ligands with Human AT_1_R

For the second set of the molecular docking calculations, we tested the binding affinity of DIZE and candesartan with AT_1_R (PDB ID: 4YAY) [33]. DIZE showed a good binding affinity (−9.67 kcal/mol) but was inferior when compared to candesartan (−11.28 kcal/mol). DIZE formed six hydrogen bonds, two with each of the following amino acids: Ser16, Arg23 and Asp281. There is also a hydrophobic interaction with Ile172 and a π-stacking interaction with Tyr92. Candesartan formed three hydrogen bonds, one with Tyr35 and two with Arg167, but also five hydrophobic interactions with Phe77, Leu81, Ile288 and two with Val108. Lastly, candesartan formed a salt bridge with Arg167 (Table 5).

In general, these ligands provided optimistic results due to their structures. The central proton and animinium protons of DIZE form strong hydrogen bonds, mostly with anionic carboxyl groups. The phenyl rings form hydrophobic interactions, as well as π-stacking interactions. In the case of the candesartan, the nitrogen of both 1H-imidazole and tetrazolide form strong hydrogen bonds. In addition, the presence of phenyl rings and the ethoxide group contribute to the formation of hydrophobic interactions. Lastly, candesartan’s anionic carboxyl group seems to form salt bridges with positively charged amino acids (i.e., Arg and Lys). Three-dimensional representations showing the interactions of DIZE and candesartan with the above-mentioned receptors using PyMOL are depicted in Figure 3.

## 4. Discussion

### 4.1. DIZE Is a Treatment Target for Hypertension and Possibly COVID-19

This study reveals that DIZE is able to reduce and inhibit AngII-mediated vasoconstriction responses in rabbit iliac arteries. DIZE has been shown to reduce AngII by activating and upregulating ACE2 activity, the enzyme responsible for converting the pathogenic octapeptide AngII into the beneficial heptapeptides Ang(1–7) and alamandine. Treatment with DIZE has been shown to reduce the blood pressure and prevent the initiation of hypertension-induced cardiac hypertrophy in rats with renovascular hypertension through the activation of MAS R and increased release of nitric oxide [11]. Additionally, the treatment of DIZE in rats with ischemia-induced myocardial infarction resulted in increased ACE2 activity, activation and mRNA expression and a reduced expression of AT_1_R [19]. Thus, suggesting the potential ability of DIZE to exert its antihypertensive abilities by directly associating with AT_1_R, which may be, in part, due the azene segment of DIZE sharing a similar structure to that of ARBs. Lastly, streptozotocin-induced diabetic rats treated with DIZE displayed the restoration of glomeruli ACE2 expression and increased AT_2_R and Ang(1–7) and reduced AngII levels in renal tissue, highlighting the possibility of DIZE directly acting on the alternative axis of the RAS [17]. The results from this study demonstrate the ability of DIZE to act as an ARB, as the rabbit iliac artery rings incubated with DIZE had abolished vasoconstriction responses to AngII, which was comparable to the vasoactive responses of candesartan (Figure 1 and Table 1). Taken together, the results from this study and others demonstrated that DIZE’s beneficial and cardiovascular protective effects may be due to direction interactions with the counterregulatory RAS axis while reducing the deleterious to classical axis, making it a new therapeutic candidate for cardiovascular diseases.

As previously reported, BV6(K^+^)_2_, a bisbiphenyl alkylated V8 bisartan that contains both imidazole nitrogens alkylated, is also an AngII inhibitor [47,48]. The sartan family has previously been identified as a class of ARB pharmaceuticals that act as antagonists of the AT_1_R by reducing AngII-mediated vasoconstriction. Furthermore, bisartan in silico docking and molecular dynamics studies have been found to exhibit a higher binding affinity for the ACE2/spike protein complex (PDB 6LZG) compared to all other known sartans [49]. DIZE contains an azene segment, homologous to the cyclic tetrazole of ARBs, forming a salt conjugate with N-acetyl glycine (aceturic acid). The extended aromaticity in the molecule of DIZE increases the acidity of the azene proton, creating a soft nucleophile for receptor binding. In particular, the triazene proton of DIZE, as the tetrazole proton in ARBs, is acidic, forming a soft base for the nucleophilic reactions at the receptor. The weak basicity is due to the delocalization of the nitrogen electron pair due to the extended aromaticity in the triazene molecule. Although the mechanisms of DIZE have yet to be fully elucidated, recent animal studies have shown that it has various beneficial effects to reduce the mean arterial pressure, liver injury and cardiac fibrosis and restores vascular dysfunction by directly eliciting vasorelaxation [11], similar to ARBs. In silico studies revealed DIZE’s strong affinity for RBD/ACE2, which is equivalent to other ARBs that have shown protective effects in COVID-19 patients in clinical practice [50,51,52,53,54,55,56,57,58,59,60,61,62,63,64]. Figure 4 shows the structure of diminazene and of the two N-Acetyl glucine molecules, which afford the diminazene diaceturate salt. DIZE has been found to activate ACE2, which breaks toxic AngII at the C-terminal amino acid, leading to aspamandine and, upon decarboxylation, to alamandine, both beneficial peptides in the RAS. Figure 5 shows sartans and vivartan, all of which bear a negative charge as a requirement for activity (tetrazolate and/or carboxylate) and, in particular, chemical structures of: losartan (EXP3174), olmesartan, valsartan, irbesartan, candesartan, azilsartan, telmisartan and eprosartan. These structures are the result of pioneer research on the RAS, which has led to the discovery of the first orally active nonpeptide ARB losartan, with EXP3174 as the active metabolite [58,59]. Five of the eight ARBs share a common biphenyl-tetrazole scaffold structure. Azilsartan contains a biphenyl-oxadiazole scaffold, which is an acidic group as a tetrazole. Telmisartan and eprosartan do not belong to the class of biphenyl-tetrazole, as the active pharmacophoric group is carboxylate not tetrazolate (Figure 5). Sartans have been recently found to protect COVID-19 patients, and this article discusses the plausible interactions of DIZE and ARBs with the basic cavity loop of the fusion region as a COVID-19 treatment target. DIZE has yet to be evaluated in COVID-19 animal models; however, we [60,61,62] and others [63,64] have advocated for its investigation as a potential treatment due to its ability to protect the cardiovascular and pulmonary systems and reduce dysfunction by targeting RAS components. This study provides evidence that the further investigation of DIZE in relevant COVID-19 animal models is required to determine DIZE’s systematic properties during SARS-CoV-2 infection. All sartans bear a negative charge (tetrazole and/or carboxylate), which allows them to complex with AT_1_R [65]. Furthermore, the sartan V8 was designed and synthesized with the reorientation of the alkyl group and hydroxymethyl group in losartan. Losartan and V8 were bi-biphenyl alkylated in an efficient method, providing a new class of ARBs called bisartans with both imidazole nitrogens to be alkylated by biphenyl tetrazole [47,48,66,67,68,69,70]. Bisartans bearing two tetrazole groups were found to be strong inhibitors of AngII in several animal models (unpublished). Figure 6 shows the chemical structures of repurposed drugs imatinib, nilotinib, ivermectin, Compound 21, AVE0991, colchicine, Molnupiravir and Paxlovid (PF-07321332), which are effective against SARS-CoV-2 [71,72,73,74,75].

### 4.2. The Bioactive CRS Conformation of AngII Triggers Hypertensive Activity and Cytokine Storm in COVID-19

Recent studies on the AngII conformation and the binding to its receptors directly implicate the cytokine storm in COVID-19 patients with the overexpression of AngII due to RAS system dysfunction [76,77,78,79,80]. An important beneficial function of ACE2 is the degradation of pathogenic AngII to beneficial heptapeptides Ang(1–7) and alamandine (decarboxylation of Ang(1–7)). This function is a pivotal link between the ACE2 deficiency and SARS-CoV-2 infection [1,4,81,82]. Our previous studies on AngII (mechanism of action and rational design of ARBs) have revealed that AngII acts at AT_1_R through a CRS, analogous to serine protease residues Tyr-His-Phe and C-terminal carboxylate [83]. CRS forms a tyrosine hydroxylate, which binds to the AT_1_R to elicit its vasoconstrictive effect [67,81,82,84], which is prevented by ARBs. The methylation of tyrosine hydroxyl led to the discovery of sarmesin, an AT_2_R antagonist confirming, together with fluorescence lifetime studies, the CRS activation mechanism [85,86,87]. Furthermore, extensive structure activity, nuclear magnetic resonance and fluorescence studies revealed the importance of the N-terminal domain and of the proline residue to stabilize the active conformation of AngII [88,89]. Our group was the first to synthesize cyclic analogs of AngII in our research to confirm CRS [68,69,70] and now appear to be candidate drugs for upregulating ACE2. Figure 7 shows the CRS of AngII and of the serine proteases. A hypothetical mechanism of action for DIZE similar to AngII and EXP3174 has been suggested.

The broad role of AngII as a potent modulator of the immune system has been demonstrated by inflammation and immunity [85,90,91,92]. Pioneer research by Steinman and collaborators has shown that blocking ACE-2 by ARBs as sartans induces potent regulatory T cells and modulates the Th1 and Th17-mediated potency [91]. This study is a guide for investigating ARBs and peptide mimetic drugs as possible regulators in the immunotherapy of autoimmune diseases [91,92].

### 4.3. The Triazene Proton Is Less Acidic Compared to the Tetrazole Proton of ARBs

The triazene part of the DIZE molecule -NH-N=N- reacts as a weak acid and proton donor, leaving behind a mild nucleophile for possible receptor binding. The triazene proton is weakly acidic compared to the tetrazole proton, which is highly acidic (pK 6). The resonance of the triazene group with the two-neighboring phenyl amidine groups creates a weak acid, which can possibly protonate a basic atom bearing a lone pair of electrons such as nitrogen and oxygen. The two terminal amidines, NH=C(pheny)-NH_2_, of the DIZE molecule react as a base and proton acceptor similarly to the imidazole of ARBs and the guanido group of arginine in AngII. This amphoteric capacity of DIZE allows the molecule to react as an acid and base. In the case of the SARS-CoV-2 spike protein, DIZE may react with acidic- and basic-binding sites targeting either linoleic acid [93] or the multi-arginine basic cavity loop of the spike protein (681–686) [94,95,96,97].

### 4.4. Tetrazole of ARBs and Triazene of DIZE Have a Common Structural Feature: The Azene

It has been suggested that DIZE exerts its antihypertensive action by reducing AngII. Reduction of the mean arterial pressure in spontaneous hypertensive rats and mice ileum arteries suggests a direct effect of DIZE on AT_1_R. DIZE and ARBs possess common structural features, which are: (i) triazene for DIZE and homolog tetrazole for ARBs, (ii) phenyl for DIZE bearing acidic triazene and phenyl for ARBs bearing acidic homolog tetrazole or carboxylic and (iii) an acidic proton on the triazene portion of DIZE and on tetrazole or on carboxylic acid for ARBs. The acidity provides nucleophiles for binding with the receptor and (iv) electrophile amidines for DIZE and imidazole nitrogen for ARBs, as both are proton acceptors. To determine the ability of DIZE to inhibit AngII-mediated vasoconstriction, a pilot study was conducted in rabbit iliac arteries. DIZE inhibited vasoconstriction as a candesartan (Figure 1). Figure 8 shows plausible interactions of (a) DIZE azene NH, (b) AngII tyrosinate and (c) ARB carboxylate with AT_1_R, reminiscent of the CRS interaction in AngII.

### 4.5. The Amphoteric Nature of DIZE May Allow Binding to Acidic and Basic Sites of SARS-CoV-2

Recently, it has been reported that the RBD of SARS-CoV-2 tightly binds the essential free fatty acid linoleic acid in three composite binding pockets. Linoleic acid binding stabilizes the locked spike protein of SARS-CoV-2, providing a treatment target [93,96]. The role of linoleic acid in stabilizing the spike protein prompted us to investigate in silico the possibility of an interaction between DIZE and the spike protein of SARS-CoV-2 through the interaction of linoleic acid with DIZE as a strategy to block the entry of virus to ACE2. To explain the possible interaction of DIZE with linoleic acid of the locked spike protein, we need to consider the nature of the chemical structure of DIZE. DIZE is a polar small di-amidine azene amphoteric molecule, which can react as acid and as a base. It reacts easily with aceturic acid (the N acetyl derivative of glycine) and is formulated as its aceturate salt [98,99]. Figure 9 shows the interaction between DIZE and the linoleic acid of RBD SARS-CoV-2 as a treatment target. The interaction of linoleic acid (carboxylate negatively charged) with DIZE (carboxymidine positively charged) should destabilize the SARS-CoV-2 S-protein, preventing its entry via ACE2. Figure 10 shows the hypothetical chemical reaction between the linoleic acid of SARS-CoV-2 RBD with DIZE [80,93].

### 4.6. DIZE as a Base Might Bind to Acidic Sites of the Spike Protein RBD/ACE2 Complex

The activation of ACE2 by DIZE most likely involves a binding site different from the binding site for AngII and is probably not specific. DIZE is a di/tribasic compound, and its protonated form is likely to bind to negatively charged molecules of all kinds. On the face of it, DIZE is likely to bind to AngII and its receptor AT_1_R. The losartan metabolite EXP3174, which has two negative charges spaced appropriately, certainly would bind well to DIZE. Another possible option is that DIZE binds well to an allosteric-binding site, different from AngII, which upregulates the formation of beneficial Ang(1–7). It is possible that this allosteric-binding site might also downregulate the binding of the SARS-CoV-2 spike protein with ACE2. Figure 11 shows Global docking of DIZE (+2e charge) to the SARS-CoV-2 RBD/ACE2 complex 6LZG.

### 4.7. DIZE and ARBs Could Bind to the Basic Cavity Loop of SARS-CoV-2 and Block the Entry to ACE2 in the Lungs

The multi-basic cleavage site P681R682R683A684R685S686 of the S1 subunit is essential for fusion of the virus with ACE2 and, thus, for infection and is an inhibition target. This basic cavity loop is reported to be the critical sequence in the S1/S2 cleavage site of the virus, which is between R685/S686 amino acids. Subunit S1 contains the RBD, and subunit S2 contains the fusion peptide. The spike protein of SARS-CoV-2 harbors the special S1/S2 furin recognizable site, entry to ACE2, which is proteolytically sensitive and a treatment target. It is of note that the globally prevailed P681H (Proline681Histidine) mutation is within the fusion region [100,101,102]. Proline is a rigid amino acid, which changes the direction of a peptide sequence, with the restricted ability to participate in charge networks, as it lacks polar groups. On the contrary, histidine is a polyfunctional amphoteric amino acid, which can behave as an acid and base and, further, can form π–π interactions with aromatic residues. This allows tighter binding as histidine with neighboring charged groups or with aromatic residues and participation in CRS, as in AngII and serine proteases.

Mutation P681R in the cleavage site 681–686 (PRRARS), detected in the new fatally spreading globally Delta variant, increases the number of arginines (RRRARS), further facilitating cleavage. Sartans, through their negative tetrazolate groups, have an additional receptor binding target, forming salt bridges and, thus, blocking cleavage and, consequently, infection. Moreover, in silico global docking studies have shown that ARBs and DIZE bind to the spike protein RBD/ACE2 complex (Figure 3), with the latter having a greater affinity than EXP3174, valsartan and eprosartan (Figure 10). Figure 12 shows a tentative interaction of multi-basic cleavage site multi-arginines (positively charged) with losartan carboxylic acid (negatively charged) and ARBs. ARBs bear a negative charge (tetrazolate and carboxylate) and may target the positive cavity loop of the spike protein, blocking its entry to ACE2 [94]. This applies to all ARBs, as they all bear negative charges. Figure 13 shows the structures of olmesartan, the active metabolite of olmesartan, and the ZD7155 AngII antagonist. Interactions between AT_1_R and olmesartan were determined by the crystal structure. The critical interactions of olmesartan are with R167, W84 and Y35 residues of AT_1_R (Figure 13A) [33,65]. The intra-CRS interactions in AngII are shown in Figure 13 [67,82].

The anchoring of olmesartan occurs through the interaction of negative tetrazole with positive Arg^167^ on the receptor, as well as with Tyr^35^ and Trp^84^. Arg^167^ formed extensive networks of hydrogen bonds and salt bridges with the carboxyl group on the imidazole moiety of olmesartan and bridges with acidic tetrazole. Tyr^35^ and Trp^84^ formed additional hydrogen bonds and π–π interactions, both with the imidazole moiety of olmesartan. Thus, imidazole of olmesartan is anchored by all three neighboring Arg^167^, Tyr^35^ and Trp^84^ residues of AT_1_R in a tight complex. The interaction of Tyr^35^ of AT_1_R with the imidazole of olmesartan is reminiscent of the interaction of Tyr hydroxyl with the His imidazole group in CRS of AngII. The mutation of receptor Tyr^35^ to alanine abolished both AngII and olmesartan binding, which shows the importance of Tyr^35^ hydroxyl for the interaction with the histidine ring of olmesartan, denoting that the AT_1_R Tyr^35^/ARB imidazole carboxylate, anchored by Arg^167^/Tyr^35^/Trp^84^ residues of AT_1_R, is a critical interaction in the CRS, analogous to the CRS in AngII. Figure 9 illustrates the chemical structures of olmesartan, the active metabolite of olmesartan and AngII antagonist sartan ZD7155 [33,65].

Similar interactions between sartan tetrazole with guanidino groups of SARS-SoV-2 spike proteins were also noted in recent in silico studies by Ridgway et al., where bisartans, bearing two tetrazoles, bound to the positive arginine guanino group and/or spike Zn^2+^ metaloproteases [49,103].

### 4.8. Physical Chemical and Toxicity Properties of DIZE

According to pkCSM and preADMET, diminazene has a molecular weight < 500 g/mol, hydrogen bonding donors ≤ 5, number of hydrogen bonding acceptors < 10 and lipophilicity less than 5. As a result, it obeys Lipinski’s Rules of Five. Since the lipophilicity is less than 5, it can be easily absorbed from the body. Additionally, Veber’s Rule is qualified because the rotatable bonds are less than 7. The same conclusions apply for aceturic acid (Table 6).

According to preADMET, the BBB values are less than 1 (Table 7). As a result, both compounds are classified as inactive in the central nervous system. The values for human intestinal absorption are high for both compounds, and this signifies that these compounds may be better absorbed from the intestine following an oral administration. Additionally, the Caco2 permeability is less than 10 × 10^−6^, and as a result, both compounds have gastrointestinal absorption ≤20%. According to pKCSM, DIZE might have the ability to penetrate the blood–brain barrier and, thus, is a potential drug for the central nervous system. DIZE has plasma protein binding more than 0.9, and as a result, it cannot traverse cell membranes or diffuse. Aceturic acid is not an inhibitor to CYP isoenzymes [104,105] and, therefore, is not toxic and does not exert other unwanted adverse effects in contrast to DIZE. These enzymes are important for the metabolism of many medications, and any inhibition of them can cause drug–drug interactions, resulting in unanticipated adverse reactions or therapeutic failures. Finally, both compounds have a relatively high skin permeability, and as a result, they favor transdermal drug delivery.

According to pKCSM, both compounds have not been predicted to be hepatotoxic (Table 8). In addition, AMES [104,105] toxicity is negative, and this means that aceturic acid is not mutagenic, in contrast with DIZE, which has been predicted to be mutagenic. Finally, DIZE is an inhibitor of HergII, and as a result, it may cause cardiotoxicity, and according to pkCSM, both compounds are not likely to be associated with skin sensitization.

## 5. Conclusions

The accumulated experimental data, the structural features that relate to DIZE, ARBs and AngII, together with the positive clinical effects from the use of ARBs in the treatment of COVID-19, provide the fundamental basis that DIZE and ARBs may be target compounds for COVID-19 therapy. DIZE and ARBs possess common structural features, which mechanistically justify a CRS interaction with their receptors. Further research is required to confirm the use of DIZE as a potential therapeutic drug for hypertension and COVID-19, as well as its ability to block the entry of SARS-CoV-2 into other host receptors [62,106].

## Figures and Tables

**Figure 2 biomedicines-10-01731-f002:**
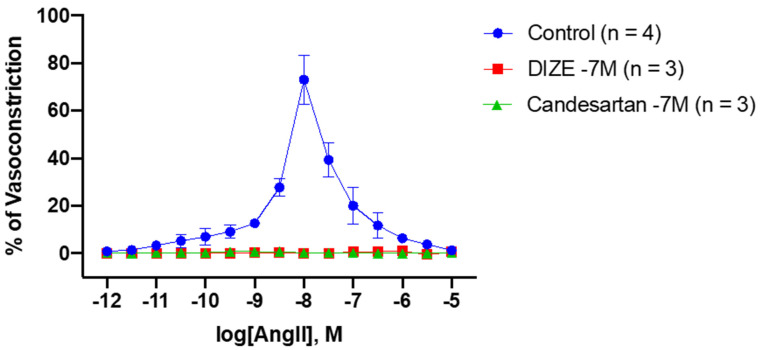
**Inhibitory effect of DIZE and candesartan during AngII-induced vasoconstriction responses in rabbit iliac arteries.** To determine the ability of DIZE to inhibit the vasoconstriction responses to cumulative doses of AngII, the iliac rings were incubated with DIZE and compared to the candesartan and control rings. As expected, incubation with candesartan, a known ARB, potently inhibited the vasoconstriction responses to AngII at doses AngII (10^−9.0^ M) to AngII (10^−6.5^ M) (mean ± SEM is shown, the significance presented in Table 1). Similar results were observed in rings incubated with DIZE at doses AngII (10^−9.0^ M) to AngII (10^−6.5^ M) (mean ± SEM is shown, the significance presented in Table 1).

**Figure 3 biomedicines-10-01731-f003:**
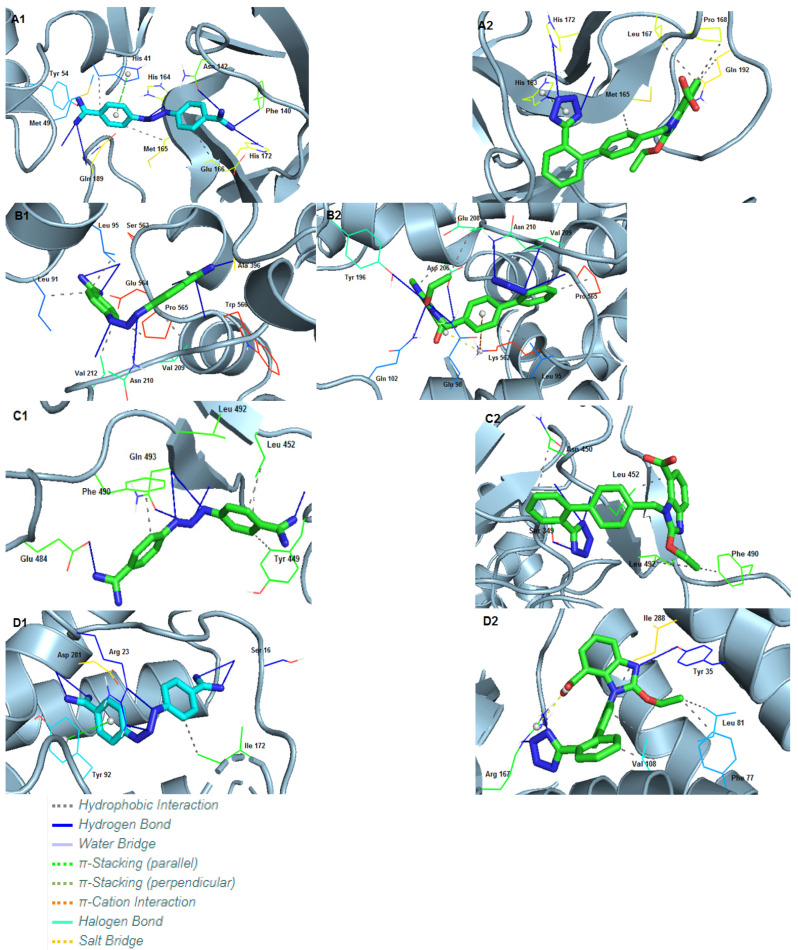
**Crystallographic grids of DIZE and candesartan incorporation into the cavity of the SARS-CoV-2 spike protein.** (**A1**) DIZE and (**A2**) candesartan incorporated into a cavity of Mpro of SARS-CoV-2 (PDB ID: 6LU7) using a crystallographic grid. (**B1**) DIZE and (**B2**) candesartan incorporated into a cavity of spike protein RBD (PDB ID: 6LZG) of SARS-CoV-2 using a crystallographic grid. (**C1**) DIZE and (**C2**) candesartan incorporated into a cavity of spike protein RBD (PDB ID: 7DDN) of SARS-CoV-2 using a crystallographic grid. (**D1**) DIZE and (**D2**) candesartan incorporated into a cavity of AT_1_R (PDB ID: 4YAY) using a crystallographic grid.

**Figure 4 biomedicines-10-01731-f004:**
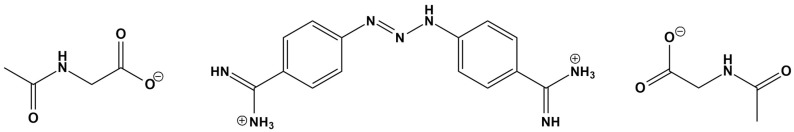
Diminazene reacts with two N-Acetyl glycine (aceturic acid) molecules to afford a salt Diminazene diaceturate. The azene hydrogen is mildly acidic due to the extended aromaticity, which delocalizes the electron pair of nucleophilic nitrogens.

**Figure 5 biomedicines-10-01731-f005:**
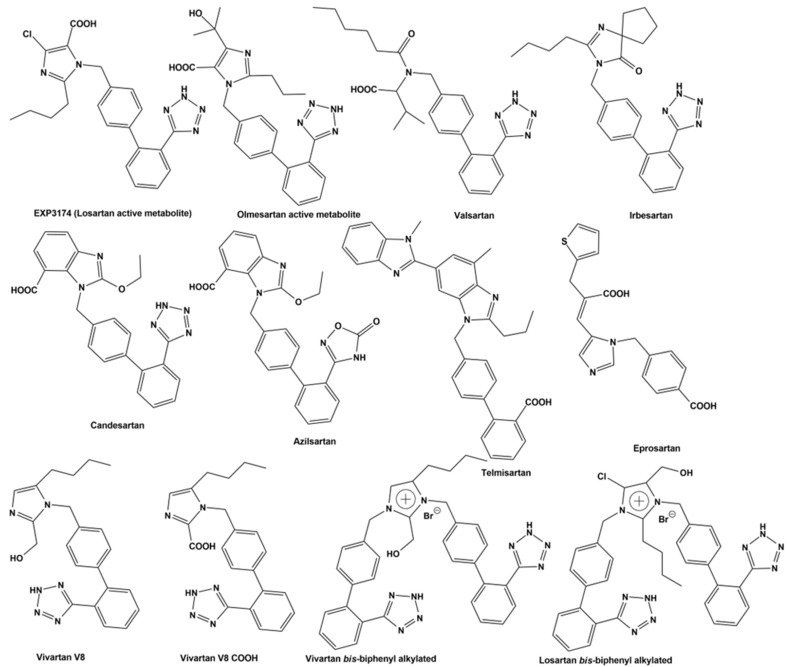
**Structures of sartans. Chemical structures of losartan (EXP3174), olmesartan, valsartan, irbesartan, candesartan, azilsartan, telmisartan and eprosartan**. Five of the eight ARBs share a common biphenyl-tetrazole scaffold structure. Azilsartan contains a biphenyl-oxadiazole scaffold. Telmisartan and eprosartan do not belong to the class of biphenyl-tetrazole ARBs. All sartans bear a negative charge (tetrazole and/or carboxylate) [57,76]. Compound V8 was designed and synthesized with the reorientation of the alkyl group and hydroxymethyl group in losartan at the laboratory of PML (Peptide Mimetic Limited) in Canada in the 1990s. The synthesis of V8 was optimized in the laboratory of Eldrug in Patras Science Park, Greece. Vivartan with similar activity as losartan did not reach clinical trials after its synthesis in 1993, as it was not cost-effective at that time. Currently, the synthesis of vivartan is efficient. Losartan and V8 were bi-biphenyl alkylated in an efficient method, providing a new class of ARBs with both imidazole nitrogens to be alkylated by biphenyl tetrazole [66,77].

**Figure 6 biomedicines-10-01731-f006:**
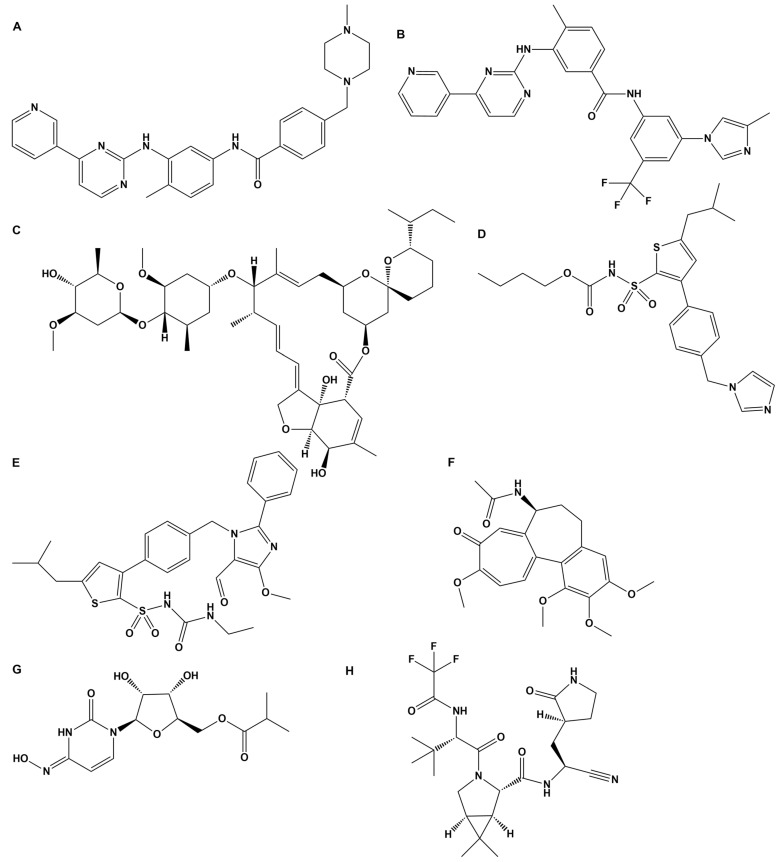
**Repurposed and antiviral drugs in the treatment of COVID-19.** (**A**) Chemical structure of imatinib. Imatinib is a specific tyrosine kinase receptor inhibitor that is used in the therapy of Philadelphia chromosome-positive chronic myelogenous leukemia and gastrointestinal stromal tumors. In silico studies have shown that losartan and imatinib are effective against SARS-CoV-2. Its functional groups are NH and NH amide between aromatic rings and may work in a similar way to DIZE [71]. (**B**) Chemical structure of nilotinib. Nilotinib is an improved analog of imatinib acting at the receptor in a similar way to DIZE. (**C**) Chemical structure of ivermectin. The FDA-approved drug ivermectin inhibits the replication of SARS-CoV-2 in vitro. The functional groups of ivermectin are the three hydroxyl groups, which may act at the receptor level through their hydroxylates [72]. (**D**) Chemical structure of Compound 21. Compound 21 is a nonpeptide agonist of heptapeptide Ang(1–7) selective for AT_2_R and has potential therapeutics for SARS-CoV-2. The functional groups are: (i) NH in between ester and sulfoxide groups attached to the thiophene and (ii) the imidazole group attached to the phenyl group of the template phenyl-thiophene [73]. (**E**) Chemical structure of AVE0991. AVE0991 is a nonpeptide agonist selective for MAS R, contributing to vasodilation of the RAS system. The functional groups are: (i) NH among the amide and sulfoxide groups attached to thiophene and (ii) the aldehyde group attached to the imidazole ring. It has the structural characteristics of other ARBs. (**F**) Chemical structure of colchicine. Several early studies have evaluated the benefit of colchicine in COVID-19 patients. Functional groups are: (i) amide NH-CO attached to the hepta ring and (ii) a carbonyl group attached to the aromatic hepta ring. Binding to the receptor may occur through NH of the amide group. (**G**) Chemical structure of molnupiravir. Merck’s first oral antiviral drug against SARS-CoV-2 infections was approved by the FDA. (**H**) Chemical structure of PF-07321332. Pfizer’s antiviral drug acts as an orally active 3 CL protease inhibitor. It binds directly to the catalytic cysteine (Cys 145) residue of the cysteine protease. It is a tripeptide mimic suitable for oral administration [74,75].

**Figure 7 biomedicines-10-01731-f007:**
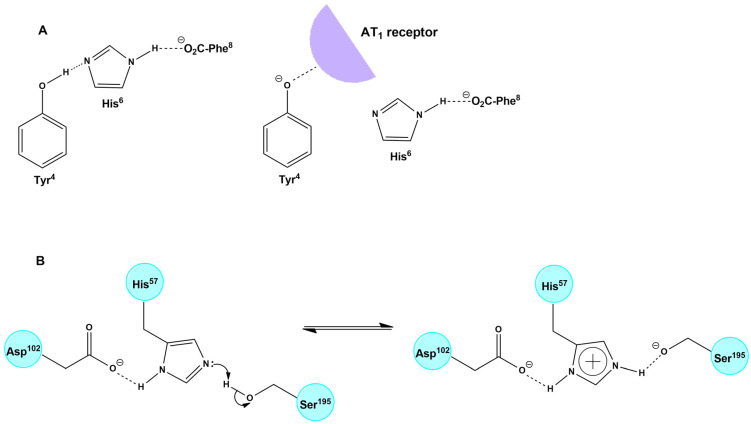
Hypothetical mechanism of action for DIZE may be similar to AngII and EXP3174. (**A**) Interaction of ANGII with the receptor through CRS (**B**) Mechanism of action of Serine Proteases: Catalytic Triad.

**Figure 8 biomedicines-10-01731-f008:**
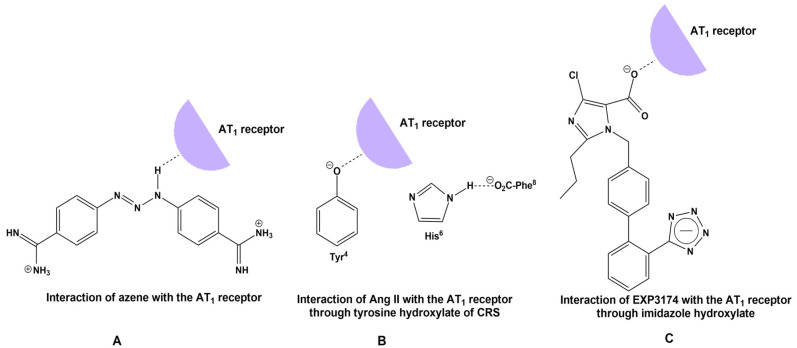
(**A**) Hypothetical interaction between azene and the receptor paralleled with the interaction of tyrosine hydroxylate of AngII. (**B**) Interaction of AngII with the receptor through CRS. A charge relay system of AngII demonstrates the interaction between the Tyr^4^ and His^6^ side chains and the C-terminal carboxylate of AngII and results in tyrosinate formation and receptor activation [82]. (**C**) Mechanism of action of Angiotensin II antagonist EXP3174 with its receptor [83].

**Figure 9 biomedicines-10-01731-f009:**
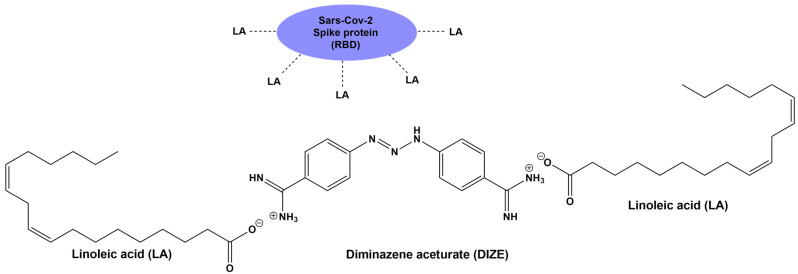
An interaction between DIZE and the linoleic acid of RBD of SARS-CoV-2 is a treatment target. The interaction of linoleic acid (carboxylate negatively charged) with DIZE (carboxymidine positively charged) could destabilize the SARS-CoV-2 spike protein. This implies that ACE2 could not bind to SARS-CoV-2 and could retain its molecular integrity, keeping its ability to degrade toxic AngII.

**Figure 10 biomedicines-10-01731-f010:**
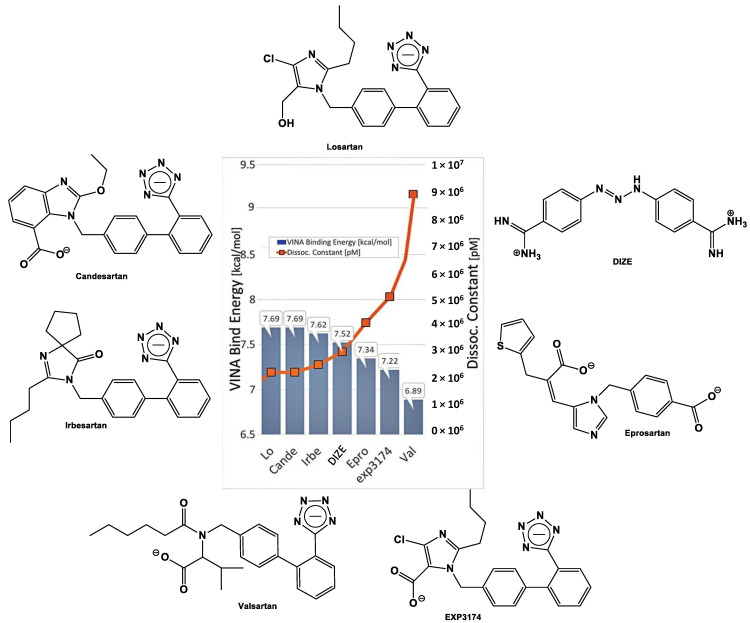
**Docking of DIZE and ARBs (Lo, Cande, Irbe, epro, Exp3174 and Val) to the zinc pocket of the SARS-CoV-2 RBD/ACE2 complex.** AutoDock VINA-binding energies are illustrated in the central plot with columns. Dissociation constants are represented by the orange line with square markers. Based on the VINA docking metric, the order of binding was: losartan > candesartan > irbesaratan > DIZE > Epro > Exp3174 > valsartan. Abbreviations: Lo, losartan; Cande, candesartan; Irbe, irbesartan and Val, valsartan.

**Figure 11 biomedicines-10-01731-f011:**
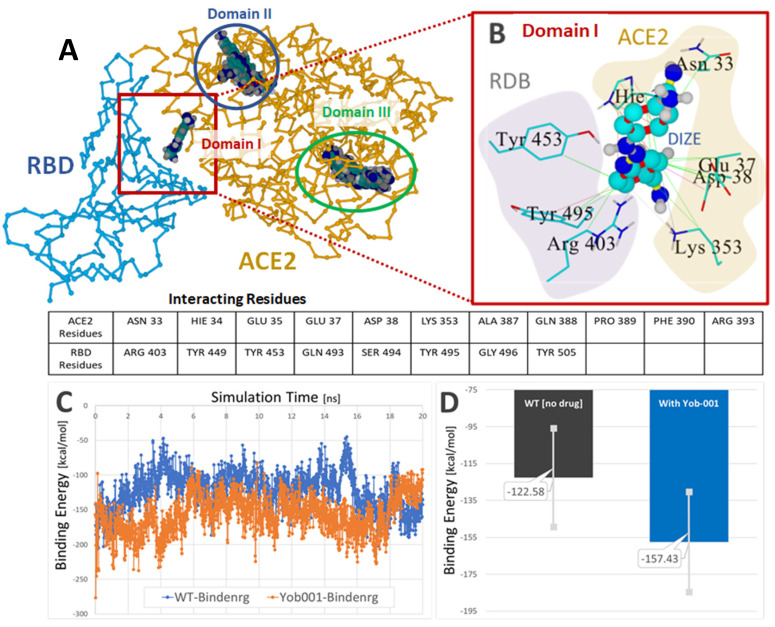
**Global docking of DIZE (+2e charge) to the SARS-CoV-2 RBD/ACE2 complex 6LZG.** (**A**) Three docking domains were identified, with DIZE positioned at the RBD–ACE interface in Domain I. The Domain I docking energy was 7.594 kcal/mol with a dissociation constant (Kd) of 2,713,464.75. (**B**) Principal interactions of DIZE in Domain I at the RBD–ACE2 interface. Hydrophobic interactions (green lines) were predominant between one aromatic ring of DIZE and RBD residues Tyr453, Tyr 495 and Arg403. Additional hydrophobic interactions for this ring involved ACE2 residues Hie34, Asn33, Glu37, Asp38 and Lys 353. Several resonant π–π interactions (red lines) involving both the RBD and ACE2 receptors helped stabilize DIZE in the interfacial pocket. (**C**) DIZE was stable in the interfacial pocket for at least 20 ns of MD simulation at 311 oK in physiological saline (0.9 wt% NaCl) at pH 7.4. (**D**) Comparison of the average RBD–ACE2-binding energies for the drug-free 6LZG complex (black bar) and the complex harboring the bound drug (blue bar) calculated over the 20-ns MD simulation. The presence of the drug resulted in a modest (approximately 34 kcal/mol) decrease in the RBD–ACE2-binding energy, suggesting DIZE may interfere with the viral attachment to susceptible host cells.

**Figure 12 biomedicines-10-01731-f012:**
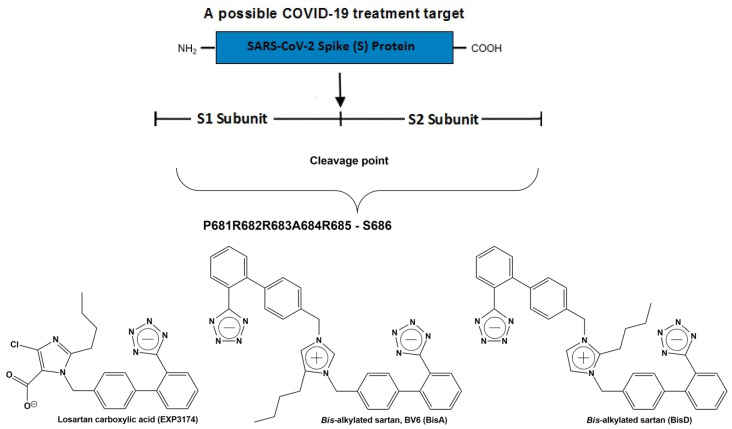
**Multi-basic cleavage site of SARS-CoV-2 is a treatment target.** Interaction of multi-basic cleavage site (multi-arginines positively charged site) with losartan carboxylic acid (negatively charged) and angiotensin receptor blockers (ARBs). ARBs bear a negative charge (tetrazolate and carboxylate) and may target the positive cavity loop of the spike protein, blocking its entry to ACE2 [94,95,96]. This applies to all ARBs that bear negative charges. Examples of ARBs are bisartans BisA and BisD bearing two tetrazoles, and based on in silico studies, this suggests that it may block ACE2 and prevent hydrolysis of the spike protein by furin and 3CL proteases triggering an infection [49].

**Figure 13 biomedicines-10-01731-f013:**
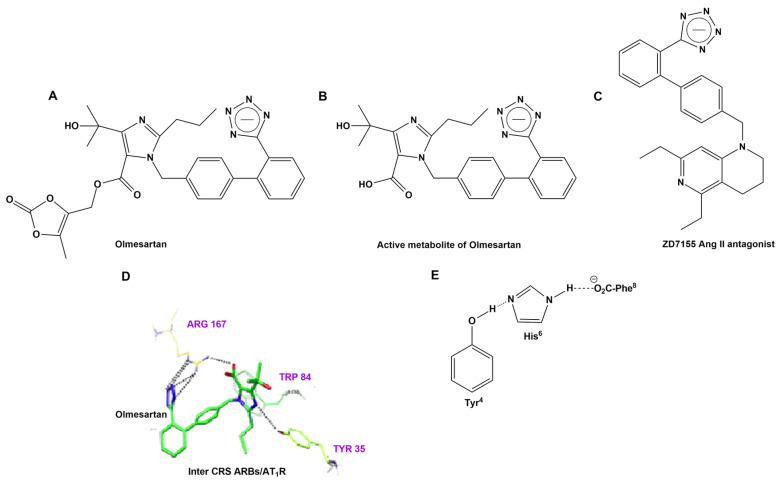
(**A**) Structures of olmesartan, (**B**) the active metabolite of olmesartan and (**C**) the ZD7155 AngII antagonist. (**D**) Interactions between AT_1_R and olmesartan were determined by the crystal structure. The critical interactions of olmesartan are with the R167, W84 and Y35 residues of AT_1_R (**A**) [33,65]. (**E**) Intra-CRS interactions in AngII.

**Table 1 biomedicines-10-01731-t001:** Significance of vasoconstriction in response to the AngII dose response between the control, candesartan and DIZE obtained from Figure 2.

Log(AngII, M)	Control vs. Candesartan	Control vs. DIZE	DIZE vs. Candesartan
−12	No significance	No significance	No significance
−11.5	No significance	No significance	No significance
−11.0	No significance	No significance	No significance
−10.5	No significance	No significance	No significance
−10.0	No significance	No significance	No significance
−9.5	No significance	No significance	No significance
−9.0	*, *p* = 0.0345	*, *p* = 0.0267	No significance
−8.5	****, *p* < 0.0001	****, *p* < 0.0001	No significance
−8.0	****, *p* < 0.0001	****, *p* < 0.0001	No significance
−7.5	****, *p* < 0.0001	****, *p* < 0.0001	No significance
−7.0	***, *p* < 0.001	***, *p* < 0.001	No significance
−6.5	*, *p* = 0.0279	*, *p* = 0.0377	No significance
−6.0	No significance	No significance	No significance
−5.5	No significance	No significance	No significance
−5.0	No significance	No significance	No significance

** p* < 0.05, **** p* < 0.001, ***** p* < 0.0001.

**Table 2 biomedicines-10-01731-t002:** Interactions of amino acid residues of the main protease of SARS-CoV-2 (PDB ID: 6LU7) with DIZE and candesartan using PLIP.

	Amino Acids of Mpro Involved and Distance of Interactions (Å)			
Type of Interaction	DIZE		Candesartan	
Hydrogen Bonds	Met49	2.15	His163	2.00
	Tyr54	2.58	Met165	3.51
	Phe140	2.11	His172	3.49
	Asn142	2.20		
	His164	2.00		
	His172	3.27		
	Gln189	3.44		
Hydrophobic Interactions	His41	3.90	Met165	3.20
	Met165	3.91	Leu167	3.82
	Glu166	3.58	Pro168	3.78
			Gln192	3.64
π-stacking (Stacking Type P)	His41	4.20	-	
π-stacking (Stacking Type T)	-		His163	3.97

**Table 3 biomedicines-10-01731-t003:** Interactions of the amino acid residues of the spike protein receptor-binding domain/ACE2 complex of SARS-CoV-2 (PDB ID: 6LZG) with DIZE and candesartan using PLIP.

	Amino Acids of Spike Protein RBD Involved and Distance of Interactions (Å)			
Type of Interactions	DIZE		Candesartan	
Hydrogen Bonds	Asn210	2.36	Gln98	2.38
	Asn210	2.26	Gln102	2.16
	Ala396	1.80	Tyr196	2.94
	Ser563	2.70	Asp206	3.70
	Ser563	2.02	Asn210	2.10
	Glu564	1.93	Asn210	2.26
	Trp566	3.03	Asn210	2.66
Hydrophobic Interactions	Leu91	3.11	Leu95	3.07
	Leu95	3.28	Asp206	3.47
	Val209	3.71	Glu208	3.81
	Val212	2.96	Val209	3.44
	Pro565	3.09	Pro565	3.48
π-cation Interactions	-		Lys562	4.94
Salt Bridges	-		Lys562	3.23

**Table 4 biomedicines-10-01731-t004:** Interactions of amino acid residues of the spike protein receptor-binding domain of SARS-CoV-2 (PDB ID: 7DDN) at the open state with DIZE and candesartan using PLIP.

	Amino Acids of Spike Protein RBD Involved and Distance of Interactions (Å)			
Type of Interactions	DIZE		Candesartan	
Hydrogen Bonds	Tyr449	1.90	Ser349	2.09
	Glu484	2.10	Asn450	2.85
	Leu492	2.09	Leu452	2.16
	Leu492	2.73		
	Gln493	2.60		
	Ser494	3.62		
Hydrophobic Interactions	Tyr449	3.78	Asn450	3.63
	Leu452	3.22	Leu452	3.41
	Leu452	3.43	Leu452	3.04
	Phe490	3.90	Phe490	3.19
			Leu492	3.82

**Table 5 biomedicines-10-01731-t005:** Interactions of the amino acid residues of AT_1_R (PDB ID: 4YAY) with DIZE and candesartan using PLIP.

	Amino Acids of Spike Protein RBD Involved and Distance of Interactions (Å)			
Type of Interactions	DIZE		Candesartan	
Hydrogen Bonds	Ser16	2.50	Tyr35	2.44
	Ser16	2.11	Arg167	2.23
	Arg23	1.93	Arg167	2.05
	Arg23	2.05		
	Asp281	2.17		
	Asp281	1.90		
Hydrophobic Interactions	Ile172	3.72	Phe77	3.25
			Leu81	3.70
			Val108	3.18
			Val108	3.44
			Ile288	3.27
π-stacking (Stacking Type P)	Tyr92	3.78	-	
Salt Bridges	-		Arg167	3.48

**Table 6 biomedicines-10-01731-t006:** The physicochemical parameters for compounds diminazene and aceturic acid.

Properties	Diminazene	Aceturic Acid
Molecular Weight	283.339 (≤500 g/mol)	116.096 (≤500 g/mol)
LogP	0.93194 (≤5)	−2.1276 (≤5)
Rotatable bonds	5 (≤7)	2 (≤7)
Hydrogen Bond Acceptors	4 (≤10)	3 (≤10)
Hydrogen Bond Donors	5 (≤5)	1 (≤5)
Surface Area	121.236 (Å^2^) < 140 Å^2^	46.501 (Å^2^) < 140 Å^2^
Water solubility	−2.952 (mol/L)	0.389 (mol/L)

**Table 7 biomedicines-10-01731-t007:** The ADME results of diminazene and aceturic acid, according to preADMET.

	Diminazene	Aceturic Acid
BBB	0.0474384 (≤1)	0.123198 (≤1)
Caco2	21.0036 (≥10 × 10^−6^ cm/s)	15.5618 (≥10 × 10^−6^ cm/s)
CYP_2C19_inhibition	Non	Inhibitor
CYP_2C9_inhibition	Inhibitor	Non
CYP_2D6_inhibition	Inhibitor	Non
CYP_2D6_substrate	Substrate	Non
CYP_3A4_inhibition	Non	Non
CYP_3A4_substrate	Substrate	Non
HIA	76.345227	65.16201
MDCK	54.7734 (≥50)	233.435 (≥50)
Pgp_inhibition	Non	Non
Plasma_Protein_Binding	9.521409 (≤0.9)	0.338226 (≤0.9)
Pure_water_solubility_mg_L	291.105	327483
Skin_Permeability	−6.65 (≤−2.5 cm/h)	−4.65 (≤−2.5 cm/h)

**Table 8 biomedicines-10-01731-t008:** The toxicity results of diminazene and aceturic acid, according to pKCSM.

Properties	Diminazene	Aceturic Acid
AMES toxicity	Yes	No
Max. tolerated dose (human)	0.456 (≤0.477) (log mg/kg/day)	1.081 (≤0.477) (log mg/kg/day)
HergI inhibitor	No	No
HergII inhibitor	Yes	No
Oral Rat Acute Toxicity (LD50)	2.802 (mol/kg)	1.829 (mol/kg)
Oral Rat Chronic Toxicity	2.714 (log mg/kg_bw/day)	2.556 (log mg/kg_bw/day)
Hepatotoxicity	No	No
Skin Sensitization	No	No

## Data Availability

Data is contained within the article.

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
