# Peer review of "Diminazene Aceturate Reduces Angiotensin II Constriction and Interacts with the Spike Protein of Severe Acute Respiratory Syndrome Coronavirus 2"

_biomedicines, 2022, doi:10.3390/biomedicines10071731_

Round 1

Reviewer 1 Report

Whereas the work seems to be carefully done, some few points need attention before publication.

1)     A theoretical strategy able to probe the conformational profile of ligands in the enzyme active site is very important. It is well-known that, from a theoretical standpoint, molecular dynamics simulations can be used to evaluate the molecular flexibility of ligands and receptors; however, it is worth mentioning that some conformational changes occur in the time scale of only dozens of nanoseconds, which could compromise the MD simulation viability for virtual screening studies, for instance. In this regard, a theoretical strategy is to select promising configurations from the docking study is crucial to determine the theoretical accuracy. In this line, I would like to suggest to authors to include a discussion about theoretical methods for selecting conformations from docking simulations, introducing the references:

BMC PHARMACOLOGY & TOXICOLOGY   Vol.: 19,   8  (2018)

Journal of Biomolecular Structure & Dynamics, v. 25, 377-386 (2008).

LETTERS IN DRUG DESIGN & DISCOVERY   Vol. 13, 360-371 (2016)

2)     In addition, the assessment of in vitro absorption, distribution, metabolism, excretion and toxicity (ADME/TOX) parameters of the potential compounds could be reported in this paper.

3)     A deeper discussion about the interaction energy and experimental results should be reported in this paper.

Author Response

1) A theoretical strategy able to probe the conformational profile of ligands in the enzyme active site is very important. It is well-known that, from a theoretical standpoint, molecular dynamics simulations can be used to evaluate the molecular flexibility of ligands and receptors; however, it is worth mentioning that some conformational changes occur in the time scale of only dozens of nanoseconds, which could compromise the MD simulation viability for virtual screening studies, for instance. In this regard, a theoretical strategy is to select promising configurations from the docking study is crucial to determine the theoretical accuracy. In this line, I would like to suggest to authors to include a discussion about theoretical methods for selecting conformations from docking simulations, introducing the references:

BMC PHARMACOLOGY & TOXICOLOGY   Vol.: 19,   8  (2018)

Journal of Biomolecular Structure & Dynamics, v. 25, 377-386 (2008).

LETTERS IN DRUG DESIGN & DISCOVERY   Vol. 13, 360-371 (2016)

Answer: Autodock provides tools that give to the user the versatility to test many conformations (we believe is better expression than configuration as configuration is related to absolute stereochemistry and here is not the case). Thus, this tool is used and among the 100 conformations generated the lowest energy structure is used for the calculations. The references are cited as suggested by the evaluator.

2) In addition, the assessment of in vitro absorption, distribution, metabolism, excretion and toxicity (ADME/TOX) parameters of the potential compounds could be reported in this paper.

Answer: ADME/TOX in silico experiments are performed by the Ph.D. student Nikitas Georgiou  who is included in the revised version among the co-authors.

3) A deeper discussion about the interaction energy and experimental results should be reported in this paper.

Answer: The docking interaction energy is not very accurate in the docking experiment.  However, it is sufficient to propose the  initiation of further experimental studies. This is what happened in our study. Indeed the rational design by the docking experiment was legitimate and the experimental results confirmed the in silico approach proving that the structural similarities of DIZE with AT1 antagonist lead to the same site of action. The experimental results did not provide any IC50 in order to have a direct comparison with in silico results. As a matter of fact this was not our purpose. The in silico results served only as a predictive tool to test DIZE for its antihypertensive properties.

Reviewer 2 Report

The authors are advised to consider the following suggestions and related queries for the suitability of its publication in Biomedicine. 

1. Highlight the hypothesis of the proposed investigation and completion of the hypothesis may be a valuable addition for human health benefits.

2. Include a literature review on a similar line of research related to diminazene aceturate that should help to highlight the current research gap that the author wants to achieve through the current investigation. It should be included in the introduction section. 

3. Literature reports various publications on diminazene aceturate as an angiotensin-converting enzyme 2 activator.  It will be interesting to highlight the novelty and significance of the current investigation compared to previously published work in the abstract and title of the manuscript.  The title should be more specific and highlight the key findings of the current investigation.

4. Methodology section required reorganization to make it more interesting to the reader.  The section "2.1.5 Statistical analysis" should be last. The ambiguity of the heading "2.1 Animal in vitro experiments". 

5. Write the purpose of In silico experiment which should connect its requirement to the outcome obtained from In vivo animal study. 

6. It is suggested to highlight the limitation of the current investigation in the discussion section and include the future directions of the study in the conclusion section.

 7. Include illustration in a form of graphical abstract which represent the whole concept of this study design including key findings and novelty of the study. 

Author Response

  1. Highlight the hypothesis of the proposed investigation and completion of the hypothesis may be a valuable addition for human health benefits.

Answer: In the introduction the hypothesis is very clear. DIZE can act as an AT1 antagonist and against COVID as it possess structural similarities with AT1 antagonists that can serve as similar pharmacophore.

  1. Include a literature review on a similar line of research related to diminazene aceturate that should help to highlight the current research gap that the author wants to achieve through the current investigation. It should be included in the introduction section.

Answer: DIZE has been postulated to exert its therapeutic abilities by modulating RAS, shifting it from a deleterious axis to one that promotes counterregulatory protective cardiovascular, renal, and pulmonary mechanisms. Animal studies have reported a positive relationship between DIZE treatment and augmented ACE2 activation and activity, as treatment with DIZE in rats has been shown to reduce blood pressure and prevent progression of renovascular hypertension-induced cardiac hypertrophy via ACE2 and MasR activation [1]. Interestingly in renal tissue, increased glomeruli and whole kidney ACE2 and AT2R expression and reduced renal AngII levels and elevated Ang(1-7) levels have been observed in male Wistar rats with streptozotocin-induced diabetes [2]. However, a combination of DIZE and PD123319 (AT2R antagonist) reversed DIZE’s beneficial effects on RAS components, suggesting that DIZE’s mechanism of action is not limited to ACE2; but rather affects different components of the ACE2/Ang(1-7)/AT2R counterregulatory RAS axis [2]. DIZE’s ability to target the traditional RAS axis is further supported by DIZE reducing ACE (60 %) and AT1R (75 %) and increasing ACE2 (30 %) mRNA levels in pulmonary tissue monocrotaline challenged male Sprague Dawley rats to induce pulmonary hypertension [3]. Additionally, treatment with DIZE in cultured human retinal pigment epithelial cells challenged with lipopolysaccharide had markedly increased mRNA and protein expression of Ang(1-7) and reduced AngII and AT1R [4]. Taken together, results from these studies suggest that DIZE may be able to target different components of the RAS in a range of tissues, of which are greatly affected by SARS-Cov-2 infection, making it an ideal candidate to further investigate its antiviral ability to be used as a treatment for COVID-19.

  1. Literature reports various publications on diminazene aceturate as an angiotensin-converting enzyme 2 activator. It will be interesting to highlight the novelty and significance of the current investigation compared to previously published work in the abstract and title of the manuscript. The title should be more specific and highlight the key findings of the current investigation.

Answer:

A new title is given to meet the reviewer’s comment.

New Title: Diminazene aceturate reduces angiotensin II constriction and interacts with the spike protein of severe acute respiratory syndrome coronavirus 2

Abstract: Diminazene aceuturate (DIZE) is a putative angiotensin-converting enzyme 2 (ACE2) activator and angiotensin type 1 receptor antagonist (AT1R). Its simple chemical structure possesses a negatively charged triazene segment that is homologous to the tetrazole of angiotensin receptor blockers (ARB), which explains its AT1R antagonistic activity. Additionally, activation of ACE2 by DIZE converts the toxic octapeptide AngII to the heptapeptides angiotensin 1-7 and alamandine, which promote vasodilation and maintain homeostatic balance. Due to DIZE’s protective cardiovascular and pulmonary effects and its ability to target ACE2 (the predominant receptor utilised by sever acute respiratory syndrome coronavirus 2 to enter host cells) it is a promising treatment for coronavirus 2019. To determine DIZE’s ability to inhibit AngII constriction, in vitro isometric tension analysis was conducted on rabbit iliac arteries incubated with DIZE or candesartan and constricted with cumulative doses of AngII. In silico docking and ligand interaction studies were performed to investigate potential interactions between DIZE and other ARBs with AT1R and the spike protein/ACE2 complex. DIZE, like other ARBs investigated, was able to abolish vasoconstriction in response to AngII and exhibited binding affinity for the spike protein/ACE2 complex (PDB 6LZ6). These results support the potential of DIZE as a treatment for coronavirus 2019.

  1. Methodology section required reorganization to make it more interesting to the reader. The section "2.1.5 Statistical analysis" should be last. The ambiguity of the heading "2.1 Animal in vitro experiments".

Answer:  We thank the reviewer for this constructive criticism.  2.1.5 was moved to the end. Methodology section was reorganized and the 2.1 heading has been changed to  “Animal model and ethics approval”.

  1. Write the purpose of In silico experiment which should connect its requirement to the outcome obtained from In vivo animal study.

Answer: The in silico experiment is used for the rational drug design. It suggests promising molecules that can evaluated in vitro and in vivo. Though, in silico studies are not performed in the real environment and may deviate from the experimental results is the only beginning tool available to start generate potential leads that forward to new more potent molecules and ultimate to drugs.

  1. It is suggested to highlight the limitation of the current investigation in the discussion section and include the future directions of the study in the conclusion section.

Answer: Future directions for DIZE are clearly described in the conclusions which highlights also the limitations of the current investigation.

  1. Include illustration in a form of graphical abstract which represent the whole concept of this study design including key findings and novelty of the study.

Answer: A graphical abstract is included in the revised version which represents the whole concept of this study design including key findings and novelty of the study.

Reviewer 3 Report

The manuscript introduced diminazene aceturate (DIZE) as dual drugs with an angiotensin type 1 receptor (AT1R) antagonistic activity and angiotensin-converting enzyme 2 (ACE2) activating activity. The authors showed DIZE’s strong affinity for RBD/ACE2 in in silico assay. And they compared new findings with obtained related findings. It is known that the receptor of COVID-19 is ACE2. DIZE has the possibility to inhibit competitively COVID-19 on ACE2. Thus, these findings will be useful for the treatment of drug development for COVID-19. Therefore, the manuscript is not too excellent to be published. In other words, the manuscript is so excellent that it should be published.

Comments

(1) Was DIZE’s affinity for RBD/ACE2 in in silico studies equivalent to the observation in vitro?

(2) Did DIZE and COVID-19 compete the same position of ACE2 in silico or in vitro?

(3) Did DIZE show protective effects in COVID-19 in in vivo assay using rodents?

(4) What is the pharmacophore for RBD/ACE2, from the findings from in silico studies and obtained compounds?

(5) In line of 29, “angiotensin type 1 receptor antagonist (AT1R)” should be replaced with “angiotensin type 1 receptor (AT1R) antagonist.”.

(6) In line of 49, “sever” should be replaced with “severe”.

(7) In line of 329, “don’t” should be replaced with “do not”.

That is all.

Author Response

(1) Was DIZE’s affinity for RBD/ACE2 in in silico studies equivalent to the observation in vitro?

Answer:  

The experimental results did not provide any IC50 in order to have a direct comparison with in silico results. As a matter of fact this was not our purpose. The in silico results served only as a predictive tool to test DIZE for its antihypertensive properties.

 (2) Did DIZE and COVID-19 compete the same position of ACE2 in silico or in vitro?

Answer:  

 Yes, it was proposed in our studies.

(3) Did DIZE show protective effects in COVID-19 in in vivo assay using rodents?

Answer:

DIZE has yet to be evaluated in COVID-19 animal models; however, we [5, 6] and others [7, 8] have advocated for its investigation as a potential treatment due to its ability to protect the cardiovascular and pulmonary systems and reduce dysfunction by targeting RAS components. This study provides evidence that further investigation of DIZE in relevant COVID-19 animal models is required to determine DIZE’s systematic properties during SARS-CoV-2 infection.

(4) What is the pharmacophore for RBD/ACE2, from the findings from in silico studies and obtained compounds?

Answer:

In silico studies suggest the tetrazole of bisartans to be the warhead pharmacophoric group which blocks the binding of RBD /ACE2 and the hydrolysis of spike protein triggering infection. Tetrazole complexes strongly with ACE2 Zn2+ and positive arginine in Delta Variant and the critical rich arginines basic cleavage site 651-656 of spike protein, preventing infection (Ridway et al CSBJ 2022, Ridgway et al  Viruses 2022).

(5) In line of 29, “angiotensin type 1 receptor antagonist (AT1R)” should be replaced with “angiotensin type 1 receptor (AT1R) antagonist.”.

Answer:

angiotensin type 1 receptor antagonist (AT1R)” was replaced with “angiotensin type 1 receptor (AT1R) antagonist.” as suggested.

(6) In line of 49, “sever” should be replaced with “severe”.

Answer:

It is replaced.

(7) In line of 329, “don’t” should be replaced with “do not”.

Answer:

It is replaced.

Round 2

Reviewer 2 Report

The authors try to improve the revised manuscript as per the given suggestions. In my opinion, it should be considered for publication upon inclusion of the editor-in-chief's opinion.